# Concept Attractors in LLMs and their Applications

## Abstract

Large language models (LLMs) often map semantically related prompts to similar internal representations at specific layers, even when their surface forms differ widely. We show that this behavior can be explained through Iterated Function Systems (IFS), where layers act as contractive mappings toward concept-specific Attractors. We leverage this insight and develop simple, training-free methods that operate directly on these Attractors to solve a wide range of practical tasks, including **language translation**, **hallucination reduction**, **guardrailing**, and **synthetic data generation**. Despite their simplicity, these Attractor-based interventions match or exceed specialized baselines, offering an efficient alternative to heavy fine-tuning, generalizable in scenarios where baselines underperform.

## 1 Introduction

Consider three distinct concepts: the Lord of the Rings universe, the Python programming language, and 19th-century romantic literature. When prompts from these concepts are given to a large language model (LLM) such as Llama 3.1 [2], we see an interesting phenomenon. For each concept, despite lexical variations among its prompts, their intermediate representations appear to collapse to distinct regions at *specific layers* – at which layer this happens varies based on the concept. For instance, prompts such as "Who is Gandalf the Grey?" and "What is the significance of Mount Doom?" share minimal similarity on the surface, yet their representations converge to nearly identical locations at layer 24. We see a similar behavior for Python-related queries such as "Help me implement a binary search tree in Python" versus "How can I

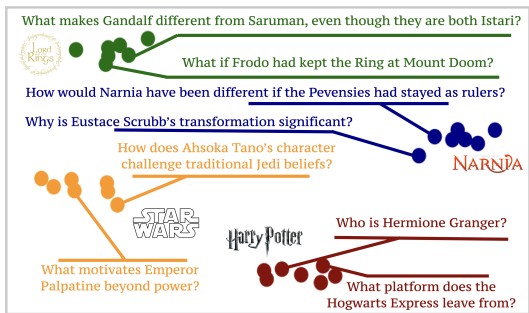

Figure 1: A t-sne[1] plot of the latent representations of Llama3.1-8B for $7 \times 4 = 28$ different prompts, seven each, for the Lord of the Rings universe, Narnia, Star Wars, and Harry Potter. Although the prompts explore different aspects of the universes and share almost no common keywords, we observe a clear clustering based on the different worlds.

find the longest non-repeating substring in Python?" and for prompts for the same genre in literature: "Discuss themes in Pride and Prejudice" and "Any easy way to recognize Byron's poetry?". Such a semantic collapse has been variously reported in some recent results. For instance, [3] notes that transformer models develop a structured latent representations that encode *belief states*. Separately, [4] suggests that due to the internal dynamics of the model, representations converge to "stable" configurations. From a more practical perspective, [5, 6, 7] showed that transformers and LLMs shape their latent space according to the underlying task. These findings, while restricted to smaller model and/or for specific contexts, cumulatively support the general idea of representation collapse.

A natural question is whether this concept-specific collapse is implied as a property of some underlying dynamical system already studied in the literature, and if so, what guidance can these existing results provide? Specifically, can we obtain strategies for important downstream use-cases? If $p_1, \cdots, p_n$ are a set of prompts related to a specific concept $\mathcal{C}$, we conjecture that the layers of our model may be acting like a dynamical system that maps semantically related inputs to proximal regions, regardless of their form at the "surface". In other words, the full sequence of layers (leading up to where the representations collapse), if viewed as a unit, implements an iterative (contractive) mapping process to an *Attractor set*, one for each concept. We will see shortly that – to the extent that our hypothesis holds – how existing results are consistent with this view of the collapse phenomena.

**Contributions.** We show that viewing the LLMs through the lens of Iterated Function Systems [8, 9] offers a meaningful (or at worst, plausible) explanation for both the layer-specific concept clustering and the subsequent generative process. The main practical benefit is that for a wide-variety of downstream tasks, which are often handled piecemeal in the literature, we can obtain a generic scheme that operates under the assumption that operating with the Attractors alone is *sufficient*. We demonstrate that careful interventions on Attractors can provide us lightweight, *training-free* solutions to a wide array of problems, from **programming language translation** and **guardrailing**, to **hallucination reduction** and **synthetic data generation**. Despite the simplicity as well as limited data/compute needs, these solutions turn out to be comparable to existing specialized approaches.

## 2 Iterated Function Systems and LLMs

There is mounting evidence that large language models (LLMs) possess emergent capabilities beyond simple rote memorization and statistical pattern matching [10]. Among the many phenomena observed in these models – from in-context learning [11] to compositional reasoning [12, 13] – we focus on a particular representation-convergence property. Our scope is specifically the collapse phenomena at specific intermediate layers. To understand this behavior through the lens of dynamical systems, we hypothesize that LLMs implicitly implement a collection of Iterated Function Systems (IFS) during forward propagation through the layers (Fig. 2).

### 2.1 LLMs implement Iterated Function Systems?

Empirically, we see that for prompts $p_i$, $p_j$ in each concept $\mathcal{C}$, there exists a layer $l$ where:

$$\lim_{l \to l_{\mathcal{C}}} \frac{1}{n^2} \sum_{i,j=1}^{n} |h_l(p_i) - h_l(p_j)| \ll \frac{1}{n^2} \sum_{i,j=1}^{n} |h_0(p_i) - h_0(p_j)| \tag{1}$$

with $h_l$ denoting the implicit transformation by the LLM up to layer $l$. This "squashing" of inter-prompt distances suggests a contractive mapping process is taking place through the layers. Our hypothesis is that this can be understood via the framework of Iterated Function Systems (IFS) [8, 9].

An IFS is defined as a finite set of contractive mappings on a complete metric space. The collective action of these mappings, defined by the Hutchinson operator [9] is:

$$\mathcal{F}(\mathbf{S}) = \bigcup_{i=1}^{N} f_i(\mathbf{S}) \tag{2}$$

and induces a compact invariant set i.e., $\mathcal{F}(\mathbf{S}^*) = \mathbf{S}^*$, which is called the Attractor of the IFS. More generally, for any initial non-empty compact set $\mathbf{S}_0 \in \mathbb{X}$, the sequence $\{\mathbf{S}_0, \mathbf{S}_1 := \mathcal{F}(\mathbf{S}_0), \mathbf{S}_2 := \mathcal{F}(\mathbf{S}_1), \cdots\}$ converges to $\mathbf{S}^*$ in the Haussdorf metric. More generally, an Attractor in a dynamical system is a closed invariant set toward which

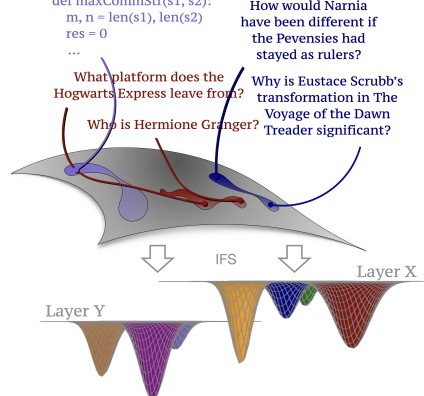

Figure 2: An LLM can be viewed as an IFS that transforms the non-linear manifold of texts into a well-behaving collection of Attractors.

trajectories from a wide class of initial conditions evolve asymptotically within its basin of attraction, and may take the form of fixed points, periodic orbits, tori, or other Attractors characterized by sensitive dependence on initial conditions [8].

Dynamical systems often exhibit Attractors—sets toward which trajectories converge. Simple systems satisfying Banach's fixed-point conditions [14] converge to a single point, while others yield more complex structures like limit cycles or strange Attractors [15]. We hypothesize that the iterative application of layer transformations in an LLM induces concept-specific invariant sets –semantic Attractors ($\mathbf{A}_l^{\mathcal{C}}$) for each concept $\mathcal{C}$– within the latent space at layer $l$. These compact regions characterize specific concepts, with convergence potentially occurring at different depths depending on the concept.

Once a sequence's representation enters $\mathbf{A}_l^{\mathcal{C}}$, it is further processed by the remaining layers and output matrix $W_{\text{out}}$ to yield a token distribution. Each Attractor may have an invariant measure $\mu_l^{\mathcal{C}}$, describing the distribution of states within it under stochastic dynamics (e.g., varied inputs aligned with concept $\mathcal{C}$). While $\mu_l^{\mathcal{C}}$ is useful for tasks like *synthetic data generation*, it does not directly define next-token probabilities in autoregressive inference, which depend on the specific input-driven state.

The attractors, $\mathbf{A}_l^{\mathcal{C}}$, are linked to the LLM's operational prefill and decode stages. During prefill, the LLM's composed layer transformations guide initial representations of an input prompt, $h_0(p)$, towards $\mathbf{A}_l^{\mathcal{C}}$, with the representation $h_l(p)$ landing within this attractor to give the initial semantic context. Then, during decode, each incremental update to the context (by newly generated tokens) is processed by these same underlying layer dynamics. For coherent generation aligned with concept $\mathcal{C}$, the evolving sequence representation at layer $l$ is continually guided towards or kept within the basin of attraction of $\mathbf{A}_l^{\mathcal{C}}$. Thus, $\mathbf{A}_l^{\mathcal{C}}$ acts like a stabilizing latent structure.

**Collage theorem.** Our operational model takes the transformation performed by the LLM for a concept and approximates it by repeatedly iterating a single affine contractive map [16], $\phi_{\text{eff}} = M_{\text{eff}}V + t_{\text{eff}}$ (with $V$ as a placeholder hidden representation), suggesting that the overall transformation, for a specific concept, can be roughly approximated by an iterated affine dynamics. We want to estimate the parameters (i.e., the matrix $M_{\text{eff}}$ and vector $t_{\text{eff}}$) and the number of

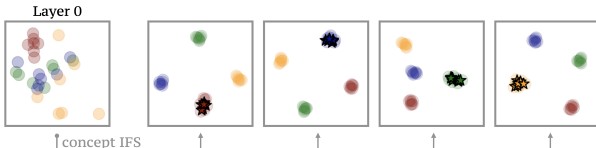

Figure 3: 4 different concepts in layer 0 (before any application of the underlying IFS, and one of the contractions of the underlying IFS we recover by solving the inverse problem for each concept separately. The circles correspond to the true vectors as obtained from the LLM in layer 24 and the stars correspond to the application of the contractions to the points in layer 0.

iterations `iter`, that best reproduce the observed mapping (Figure 2). This is achieved by minimizing the discrepancy between the LLM's observed states at the Attractor layer and the states predicted by iterating $\phi_{\text{eff}}$ from the initial prompt representations:

$$\min_{M_{\text{eff}}, t_{\text{eff}}, \texttt{iter}} \sum_{j=1}^{N} \mathcal{D}\left(h_l(p_j), \phi_{\text{eff}}^{\texttt{iter}}(h_0(p_j))\right) \tag{3}$$

subject to $M_{\text{eff}}$ being contractive (e.g., its operator norm $|M_{\text{eff}}|_{op} < 1$). We apply this `iter` times, and $\mathcal{D}$ is a suitable distance metric. This single map $\phi_{\text{eff}}$ defines a simple Iterated Function System (IFS). The unique Attractor of this 1-map IFS is its fixed point, $V^*$ to which all trajectories $\phi_{\text{eff}}^{k}(V)$ (for any initial $V$) converge as $k$ grows. The observed empirical set $\mathbf{A}^{\mathcal{C}}$ is then interpreted as the collection of states reached after `iter` applications of $\phi_{\text{eff}}$ starting from the initial set $S_0$. If, as empirical evidence for many concepts suggests, this 1-map model provides a good first-order approximation, then $\mathbf{A}^{\mathcal{C}}$ would be expected to lie in the vicinity of $V^*$. The Collage Theorem [8] states that if $\mathbf{A}^{\mathcal{C}}$ is indeed close to the true Attractor $V^*$ of our fitted $\phi_{\text{eff}}$, then $\mathbf{A}^{\mathcal{C}}$ should be well "collaged" by $\phi_{\text{eff}}$ itself; i.e., $d\left(\mathbf{A}^{\mathcal{C}}, \phi_{\text{eff}}(\mathbf{A}^{\mathcal{C}})\right)$ should be small. While the iterated single affine map is simple, for concepts whose empirical Attractors $\mathbf{A}^{\mathcal{C}}$ exhibit more complex geometries (e.g., disjoint sets or intricate fractal structures not well approximated by convergence to a single point), a richer effective IFS comprising multiple affine maps might be necessary. This would involve finding $\phi$'s and an iteration count `iter`' that minimize $d\left(\mathbf{A}^{\mathcal{C}}, \mathcal{F}^{\texttt{iter}'}(S_0)\right)$, where $\mathcal{F}$ is the Hutchinson operator for the candidate set of $\phi$'s. Alternatively, one could model the geometry of $\mathbf{A}^{\mathcal{C}}$ directly by finding an IFS whose intrinsic Attractor matches $\mathbf{A}^{\mathcal{C}}$, by minimizing the collage error. These approaches are more involved but grounded in IFS theory.

**Does this perspective add to existing results?** Several recent results have indirectly hinted at the IFS-like nature of the LLMs, and more generally transformers, for specific tasks, datasets, and

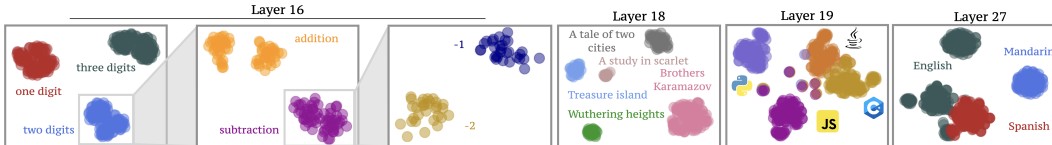

Figure 4: Attractors in Llama3.1-8B [2]. From the fractal structure of the task vectors in layer 16, to literature-based Attractors in layer 18 and programming-based in layer 19, the treatment of an LLM as an IFS allows us to recover (and use) them in multiple applications.

architectures. [4] describes how the intermediate layers of an LLM converge to different "Attractor" points/vectors as the context window of the LLM increases. The result in [17] examines the Attractors formed in the output layer of an LLM, discovering that paraphrasing results in 2-period cycles. The authors in [3] present evidence that transformers develop internal representations corresponding to "belief states" over hidden variables in the data-generating process. This phenomenon mirrors the behavior of an IFS, belief states in [3] can be viewed as specific points within concept Attractors that encode probabilistic information about possible continuations. Notice that the fractal structures reported in [3] arises naturally from known properties of IFS: systems whose repeated application to an initial set converges to a unique invariant set with so-called *self-similar* properties.

## 2.2 A preliminary investigation of Attractors

Before evaluating their practical utility, we first examine the nature of Attractors and their underlying IFS across various concepts and datasets as a sanity check.

**Induced tokens.** To understand what the Attractors represent, we average the vectors for each of the four fictional worlds from Fig. 1 to approximate their Attractor points, then project them to vocabulary space via the LLM's final linear layer. The top induced tokens (table 1) support our hypothesis, revealing meaningful associations—including tokens not present in the original texts, such as the pound symbol (£), filming locations (Auckland, NZ), or author connections (C.S. Lewis and J.R.R. Tolkien). This suggests the Attractors capture the underlying "essence" of each world, beyond surface-level content.

Table 1: Top induced tokens of Attractors.

| Concept | Tokens |
|---|---|
| Harry Potter | Harry, wizard, Hogwarts, magical, Voldemort, London, British, £ |
| Lord of the Rings | Lord, Tolkien, Middle, Auckland, NZ |
| Narnia | Kingdom, Tolkien, British, Oxford, Aslan |
| Star Wars | Imperial, Star, galaxy, Galactic, Jedi, Empire, Skywalker, Force, powerful |

**Different concepts, different layers.** While for functional worlds, as in Figure 1, we see that the LLM forms clear Attractors in layer 24, this is not the case for all families of concepts, and not discussed in many existing results. We will see later that different families of concepts form Attractors in different layers. For example, we observe the same behavior in layer 19 for programming languages, in layer 27 for natural languages, and in layer 18 for literature books (Figure 4).

**Same concept, multiple Attractors.** Previously, we modeled each concept as a single Attractor (or Concept Vector) in the LLM's latent space. However, some concepts may decompose into multiple sub-concepts. For instance, English forms two distinct Attractors when combining datasets with different semantic styles (https://www.manythings.org/anki/spa-eng.zip, https://huggingface.co/datasets/swaption2009/20k-en-zh-translation-pinyin-hsk; see fig. 4). This fragmentation is even clearer in layer 16, where tasks produce multiple Attractors based on the number of digits per example.

**A fractal-like structure in the Attractors.** In fig. 4 (left), replicating the setup from [5], we observe a fractal-like structure in the ttractors derived from simple arithmetic tasks (e.g., adding 1, subtracting 2). At a high level, Attractors cluster by the number of digits in the examples. Zooming in, subclusters emerge based on task type (addition vs. subtraction), and further divisions align with specific values being added or subtracted. This hierarchical structure aligns with theoretical findings in [3], suggesting a fractal organization of Attractors in this setting.

**LLMs and World Models.** There is much discussion related to whether LLMs operate with an explicit, internal world model [18]. Based on the empirical analysis described so far, we find that there is at least partial evidence to support the idea that the models indeed harbor a *fuzzy* understanding

of the world, which is better expressed partially across many of these intermediate layers. In the subsequent section, we will focus on how we can better exploit this fuzzy world model of the LLMs and propose practical, training free solutions to a number of use cases.

# 3 Attractor for concept detection

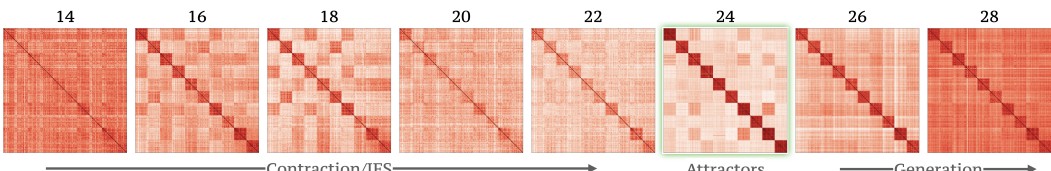

Figure 5: Cosine similarity between all prompts' from TOFU forget05 [19]. The first 20 rows/columns of each heatmap correspond to questions about the first author, the second 20 about the second author, and so on. The forming of author-based Attractors is apparent and it becomes clearer in layer 24.

Machine unlearning is a active research area, stemming from computer vision [20] where many widely used datasets include images of individuals who did not consent to their use. The training datasets of contemporary LLMs are also prompting concern about compliance with the Right to Be Forgotten [21] and similar regulations. Due to the size of these models, retraining or fine-tuning (e.g., [22, 23, 24, 25]) is often too costly. Moreover, since removal requests are continuous, efficient online unlearning is desirable. To evaluate unlearning in LLMs, Maini et al. [19] proposed the TOFU benchmark, where models must forget certain fictional authors while retaining performance on others and unrelated tasks.

**Existing solutions.** LLM unlearning methods fall into two main categories: (1) weight reversion and (2) guardrailing. *Weight reversion* seeks new parameters $\theta'$ close to those of a model trained without the forget set, $\theta^*$. Early work [26] proposed lightweight fine-tuning to forget specific content (e.g., Harry Potter), but it does not scale to frequent or multi-instance requests. Recent PEFT-based methods [27, 28] improve efficiency but still require retraining and access to retention data, making them impractical for continuous unlearning. *Guardrailing* avoids changing model weights by intervening at input/output levels. While widely used, such techniques are typically shallow and vulnerable to jailbreaking [29, 30]. Hybrid approaches like Preference Optimization [19] use gradient ascent and placeholder outputs but still involve full model fine-tuning and retention data. Other methods (e.g., [31]) inject noise using concept classifiers, offering improved efficiency but still requiring training and retention data for each concept.

**A training-free approach.** We propose a train-free concept guardrailing method for LLMs that requires only data from the concept to be removed –no retention data needed– making it both computationally and data efficient. As shown in fig. 5, certain concepts (e.g., TOFU authors) form clear attractors in intermediate layer 24. We estimate each attractor by averaging hidden activations across the concept's samples. At inference, we compute the cosine similarity between the output's

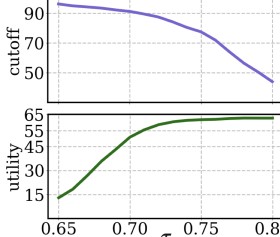

| Method | forget01 Utility ↑ | forget01 Rouge ↓ | forget05 Utility ↑ | forget05 Rouge ↓ | forget10 Utility ↑ | forget10 Rouge ↓ | Train Free | No ret. data |
|---|---|---|---|---|---|---|---|---|
| Original | 62.67 | 97.67 | 62.67 | 97.67 | 62.67 | 97.67 | - | - |
| Grad Asc | 60.24 | 43.61 | 00.00 | 00.09 | 00.00 | 00.00 | ✗ | ✓ |
| Grad Diff | 60.59 | 44.80 | 32.44 | 01.85 | 58.23 | 00.32 | ✗ | ✗ |
| Pref Opt | 62.36 | 31.31 | 47.85 | 03.27 | 53.95 | 06.02 | ✗ | ✗ |
| NPO | 45.32 | 24.27 | 17.14 | 19.68 | 17.01 | 20.10 | ✗ | ✓ |
| NPO-RT | 48.96 | 26.55 | 54.14 | 28.93 | 49.97 | 23.80 | ✗ | ✗ |
| ECO | 62.57 | 03.32 | 62.57 | 07.62 | 62.35 | 06.94 | ✗ | ✗ |
| **Ours** | 62.67 | 00.48 | 61.20 | 10.33 | 61.34 | 19.54 | ✓ | ✓ |

Figure 6: (left) Model utility and cutoff percentage as a function of the threshold $\tau$ for TOFU forget10 [19]. Model utility determines the impact that the guardrailing has on the general answering abilities of the LLM while cutoff represents the percentage of questions regarding the forget set that are detected and guardrailed. (right) Model Utility and Forget Rouge of our train-free method compared to the typical (e.g., Gradient Ascent) and most recent trainable approaches (e.g., NPO [32] and ECO [33]). Although our approach is the only train-free approach and it requires no retention data, it is better than most baselines, offering also a greater control over the tradeoff of Model Utility vs Cutoff/Rouge, with the introduction of $\tau$.

attractor and the stored one; if it exceeds a threshold $\tau$, the response is blocked and replaced with a fixed message (e.g., "I cannot provide information about author X due to removal request <id>"). This requires only a single forward pass and no training.

**Evaluation.** Figure 6 (left) shows the cutoff percentage and the model's utility for different values of $\tau$ and for all 3 versions of the TOFU benchmark [19]. We can observe that even for the hardest version (forget10), the model's utility remains high while we enjoy a cutoff percentage of more than 90%. For specifically chosen values of $\tau$, we show in Figure 6 (right) that our train-free approach is competitive with many heavier, trainable solutions. At the same time, the use of $\tau$ allows a finer control over the tradeoff of forgetting versus model utility.

# 4 Attractors for traversals

Treating the LLM as an IFS, and more generally a dynamical system, allows us to intervene on its trajectory and guide it towards specific Attractors. From a dynamical system perspective, if we assume that the LLM can be characterized from a function $f$ such that $dx/dt = f(x)$, then, given a target Attractor $y$, we can modify the system as $dx/dt = f(x) + \lambda(y - x)$ and steer it towards another Attractor $y$, with $\lambda$ being influenced by the underlying dynamics of the system (robustness to perturbations, distance of Attractors, etc.).

Such an approach, called *steering*, has been variously studied. We know that carefully chosen vectors can steer a model's behavior so that its output is less toxic, more poetic, etc. [34, 35, 36], essentially steering the model internally to different Attractors. However, many of these approaches require training the model itself or auxiliary smaller networks (e.g., [36, 37, 38]), while other works require carefully chosen data that satisfy some, more or less restrictive, assumptions (e.g., [35, 39, 40]).

Unlike methods requiring extensive retraining or retention data, we show that simply adding or subtracting Attractors at selected intermediate layers can influence LLM behavior across tasks –from detoxification to code translation– without these constraints. Surprisingly, in practice, the *before* Attractor is unnecessary, removing the need for retention data entirely. Despite requiring only a single forward pass over target data and no training, our approach matches the performance of more resource-intensive methods.

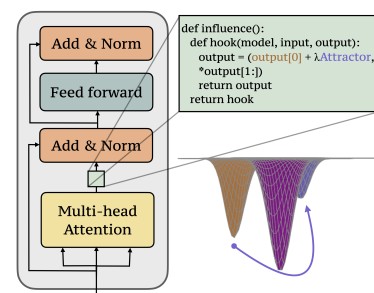

Figure 7: Influencing the dynamics of the LLM by adding the target Attractor. The only modification needed is the introduction of a forward hook on the appropriate layer.

## 4.1 Drifting away from the toxicity Attractor

| Method | Toxicity ($\downarrow$) | Rouge ($\uparrow$) | Train free | No ret. data |
|---|---|---|---|---|
| Original dataset | 84.58 | - | - | - |
| Llama2-7B | 71.60 | 73.15 | - | - |
| ICL | 66.81 | 74.19 | ✓ | ✗ |
| LoRA | 49.94 | 57.32 | ✗ | ✗ |
| ICV | 39.54 | 65.97 | ✓ | ✗ |
| **Ours** | **37.66** | 61.27 | ✓ | ✓ |

| | | | |
|---|---|---|---|
| ok - i can second this , that nicaraguan coffee is f\*cking amazing. | this sh\*t is why i moved to cali and refuse to look back. | lol sh\*t i need change. | for example i don´t give a sh\*t about your opinions , yet you still comment. |
| I can confirm that the coffee from Nicaragua is excellent. | This is why I moved to California and will not look back. | I need to change my life | I don't care about your opinions, but you still commented. |

Figure 8: (left) Toxicity score and Rouge on ParaDetox. Although our lightweight approach requires no training or even retention data, it is reducing significantly the toxicity while maintaining the textual quality. (right) Toxic examples and the modified passages according to our method.

Multiple works have shown that careful manipulation of the activations across the LLM's layers allows us to control its behavior, and a common application is toxicity reduction. We note that these ideas impose one or more restrictive requirements on the data format, such as the need for retention data, or even the existence of paired data [35, 34]. Here, we check whether the estimation of the toxicity Attractor alone allows us steer the generation away from it and thereby, reducing the toxicity content of the LLM's output. No additional assumptions on the data are needed. Using the ParaDetox [41] dataset, we obtain a single vector estimate of the toxicity Attractor on layer 16 and, then, during generation, we subtract this value from each token's activation on layer 16, essentially discouraging the generation to converge to the toxicity Attractor. Although we only require the

toxicity Attractor/vector, our targeted approach performs better than many of the existing (but more restrictive) solutions.

**Evaluation.** In Figure 8, we show that our approach, without any need for training/retention data, performs similar as ICV [35] which needs a PCA projection of the differences between paired samples. We also appear to perform better than LoRA fine-tuning or the more lightweight In-Context Learning [11]. To assess both the reduction in toxicity as well as any potential drop in the quality of the generated text, we report both Toxicity [42], as well as the Rouge score [43]. Our approach is one of the few training free methods and the only one that requires no retention data. We find that relaxing these requirements does not lead to a performance drop, instead a performance gain. Finally, we should note that there are practical benefits of our lightweight approach.

## 4.2 Switching language Attractor on the fly

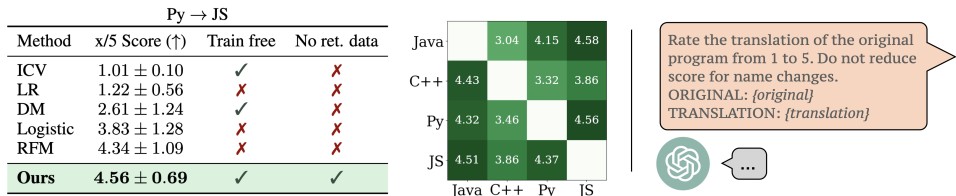

Figure 9: (left) LLM as a transpiler. For all pairs of the four considered languages, switching the Attractor to the target language can successfully make the LLM act as a transpiler without any specific such instructions or retention data. (right) Using o4 to judge the quality of the generated translations.

LLMs are extremely capable at code comprehension and composition [44, 45, 46]. Other than use as a code-generation assistant, an important use case is as a transpiler, especially for programming languages with limited support. Typically, the approach involves a data-intense stage of fine-tuning on code-specific data (e.g., [47, 48]). Some recent works have evaluated the limits of zero/few-shot transpiling in LLMs [49, 36].

As we showed in Figure 4, programming languages form Attractors on layer 19 of Llama3.1-8B. Here, we examine whether we can use these Attractors and use the LLM as a transpiler. Given only code block in an input language and without any specialized instructions, can we translate it to another, target language? Using 100 solutions of LeetCode problems in Python, Java, C++, and Javascript, we obtain an estimate of the corresponding Attractors in layer 19. Assuming that the model, provided with a solution in language X converges to the corresponding Attractor X, we examine the effect on the generation process if we traverse the Attractor space and move to the Attractor of another language Y.

**Evaluation.** To evaluate the quality of the generated code, we use o4-judge to provide us with a score of the quality of the generated code in the target language. As shown in fig. 9, we can successfully repurpose the LLM as a transpiler without any demonstrations (zero-shot) as well as no other relevant information in the prompt. We achieve impressive results for all pairs of the 4 considered languages. We do not require any retention data, additional training, or an increase in the inference time. We obtain a score better than other simple, train-free approaches (e.g., Difference of Means (DM) [36] and ICV [35]) as well as approaches that involve training auxiliary classifiers (e.g., RFM, LR [36]).

## 4.3 Remaining on the visual Attractor

Hallucinations in LLMs are a well-known challenge [54, 55, 56], and they worsen in Vision-Language Models (VLMs) due to the fading memory effect—where the model's attention to visual input diminishes during generation, "forgetting" essentially the visual input [57, 58]. We hypothesize that this results from a shift between Attractors: although VLMs initially align with an Attractor encoding visual input, the LLM backbone tends to drift toward a text-only Attractor due to its pretraining. To counter this, we propose adding the initial visual Attractor vector (computed at the first generation step) to the hidden state at each subsequent step, reinforcing visual grounding. Unlike before (fig. 7), our method dynamically calculates and maintains the visual Attractor throughout generation on each generation.

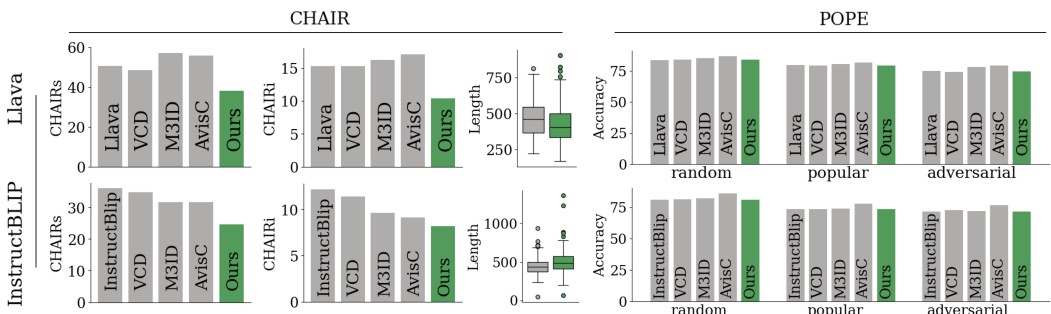

Figure 10: CHAIR [50] and POPE [51] on Llava-1.5 [52] and InstructBLIP [53]. While our approach maintains performance on the discriminative questions (POPE) it significantly reduces the hallucinations in the generative tasks (CHAIR), without affecting the length of the generated descriptions.

**Evaluation.** Compared to other train-free approaches (e.g., [57, 59, 60]), our algorithm does not lead to an increase in inference time, since it does not require multiple forward passes. Despite its simplicity, the results are strong, leading to a significant reduction in the hallucination rate of two widely used VLMs (InstructBLIP [53] and Llava-1.5 [52]), as shown in Figure 10 (CHAIR). Our modification also does not affect the general abilities of the VLM, resulting in a similar (or slightly improved) performance on discriminative questions.

# 5 Attractors perturbation for data generation

Recent works have suggested that LLMs can be used for generation of new samples, similar to a (usually small) real dataset. Multiple works have explored different ways that LLMs can be prompted successfully to generate accurate samples, as well as different multi-step ways that can further improve the quality of the newly generated samples [61]. Others have highlighted the difficulty of designing proper prompts and the tedious trial and error required, and proposed a minimal fine-tuning of an LLM so that it serves as an Autoencoder that can produce new samples via high temperature sampling [62].

**Limitations of Temperature sampling.** LLM output variability is often controlled via Temperature (and its related parameters top-K and top-P), introducing stochasticity into generation. However, even at high randomness, outputs often remain limited and lack diversity when generating text similar to existing data [63, 64, 61]. Usually, this problem is addressed by a more carefully tuned input prompt or the use of many different ones. But such an approach is not easily scalable and not applicable to tasks involving large synthetic data generation.

Specifically, a common way to address the lack of diversity (beyond the capability of temperature sampling alone) is to perform multiple forward passes with different prompts/instructions, while keeping the same sample from the original data. Several proposals show that carefully tuned instructions/prompts can help the model generate different, more diverse types of synthetic samples [63, 64, 61]. However, this requires tedious and careful design of the prompts in a non-automated way, with multiple rounds of trial-and-error in some cases. It is also problematic when we consider large, diverse datasets that do not adhere to a single "type", like the ones we will examine here (e.g., BoolQ [65]).

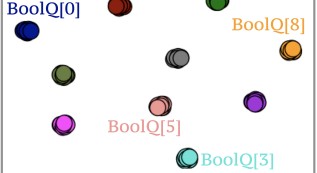

Figure 11: Sample-based Attractors for different generation instructions. Each Attractor corresponds to one sample from BoolQ[65].

## 5.1 Attractor perturbations: Replicating the effect of multiple tailored instructions

Similar to the experiments reported so far, we can examine if there are sample-wise Attractors for multiple different and diverse instructions. Is there a layer where we can observe a "collapse" based on the different samples? The answer is yes, and we show this in Figure 11. We curated a list of 10 different instructions tailored to BoolQ dataset, and we observed that on layer 16, for the same sample, all trajectories converge to the same Attractor, forming essentially sample-wise Attractors. Based on this observation, we examine whether we can replicate the effect that multiple different prompts

|  | BoolQ | | AG | |
| --- | --- | --- | --- | --- |
|  | Qwen2.5-0.5B | GPTNeo-1.3B | Qwen2.5-0.5B | GPTNeo-1.3B |
| No train | 38.47 | 38.53 | - | - |
| No augmentation | 62.54 | 62.17 | 30.66 | 23.46 |
| Temp sampling | 64.16($\pm$2.98) | 64.80($\pm$2.80) | 82.91($\pm$2.58) | 50.96($\pm$19.45) |
| **Ours** | **69.28($\pm$0.88)** | **67.77($\pm$1.86)** | **85.64($\pm$0.59)** | **72.06($\pm$7.51)** |

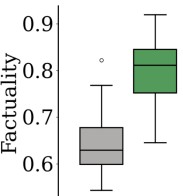

Figure 12: (left) Test-set accuracy on BoolQ [65] and AG [66] when trained with synthetic datasets generated through temperature sampling and our approach. In all cases, our dataset results in a more generalizable model with better performance. (right) Factuality of generated facts about popular figures with temperature sampling (gray) and our approach (green). We observe a more than 20% increase in the factuality on average.

have on the generated data – simply by perturbing the Attractor estimates we obtain using a single instruction. Using only a single (and perhaps simple) prompt/instructions, can we generate multiple, diverse samples without the need to increase the temperature and deal with corrupted, non-sensical samples? As we will show shortly, such simple, train-free and tuning-free approach can result in better quality data, and we check this in both direct and indirect ways.

**Estimating the quality of the generated data.** Here, we consider textual two datasets: BoolQ [65] and AG [66]. Although both datasets are relatively large/diverse, to limit the influence of the original train set on the final results and better assess each generation method, we consider a minimal version for each dataset, with only 100 samples. Based on these 100 samples, we prompt the model (Llama3.1-8B) to generate new synthetic samples, following both approaches (the typical temperature sampling and ours). One common way to assess the quality of the generated data is by fine-tuning a smaller LLM on the newly obtained version of the data [61] (called an indirect evaluation). Here, we consider Qwen2.5-0.5B [67] and GPTNeo-1.3B [68] as the small LLMs that we will finetune on the synthetic collections. In fig. 12 (left) we show the accuracy obtained on the real test set by training each model with each version of the dataset. The improved quality of our method is clear and we find that it leads to better results in all cases.

**Estimating the factuality of the generated data.** Besides the indirect comparison described above, we can also examine the quality of the generated samples directly. Following [69], we prompt the model to generate facts for a collection of randomly selected celebrities and historical figures. To evaluate the factuality of each fact, we uses o4-judge by prompting it to output true or false for each of the generated facts. In fig. 12 (right), we show that the factuality of the generated samples is much lower using temperature sampling. Using Attractors, we achieve an absolute increase of 20% on average. A detailed improvement for each person separately can be found on the appendix.

# 6 Conclusion

This work is based on the hypothesis that the evolution of hidden representations of prompts in Large Language Models (LLMs), specifically their convergence to distinct internal representations (for semantically related prompts), can be understood through the framework of Iterated Function Systems (IFS). We check that LLM layers progressively map inputs towards concept-specific "Attractors" in their latent space. Building on this perspective, we evaluated a range of simple, training-free ideas that directly manipulate these identified Attractors. On a diverse set of practical tasks, including machine unlearning (guardrailing against specific concepts), guiding LLM generation for tasks like code translation and toxicity reduction, mitigating hallucinations in vision-language models, and improving the diversity and factuality of synthetic data generation, we find that our proposal offers surprisingly strong performance. It is computationally efficient and there is no need of re-training or fine-tuning, and offers a clear and promising direction for evaluating applicability in other use-cases.

**Impact & Limitations.** A single modeling of the LLMs (as IFS) can lead to multiple solutions for very different problems. We believe that these ideas can help solve many more practical problems that LLMs face or expand their capabilities. One limitation of our work is that we require direct access to the hidden activations of the model, for both estimating and manipulating the identified "concept Attractors". Our methods cannot typically be implemented through standard API calls to LLMs, which usually only provide black-box input/output functionality. Due to compute constraints, we focused on evaluating our methods on models with up to 8B parameters but it will be interesting to check if a similar Attractor phenomena holds for much larger models.

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
