# OpenReview forum: "Concept Attractors in LLMs and their Applications"
_NeurIPS.cc/2025/Conference — Submitted to NeurIPS 2025_

### Official Review · Reviewer_iGkG · 2025-07-02

**Clarity:** 2
**Significance:** 2
**Originality:** 2
**Rating:** 4
**Confidence:** 3

**Summary:**

This paper discovers that prompts associated with the same ``concept’’ tend to converge toward an attractor in the representation space of intermediate layers, even when those prompts exhibit significant semantic differences. Moreover, different concepts are shown to converge to attractors located in different layers. This layer-specific convergence enables manipulation tasks such as unlearning and translation by directly operating on attractors—without modifying the model parameters.

**Questions:**

1. In Section 4, how does the proposed method of introducing attractors differ from the representation injection mechanism described in [1]?

2. In Figure 5, why do Layers 20 and 22 perform worse than Layers 16 and 18? Can the IFS account for this non-monotonic convergence across iterations?

3. Why is the average of vectors assumed to represent the attractor? Could this introduce bias related to prompt distribution? Additionally, does the concept of a single average vector contradict the concepts of multiple attractors?

[1]. Knowledge Neurons in Pretrained Transformers, ACL, 2022

**Ethical Concerns:**

["NO or VERY MINOR ethics concerns only"]

**Final Justification:**

The authors have addressed most of my concerns and I'd like to keep my positive rating.

**Limitations:**

yes

**Quality:**

3

**Strengths And Weaknesses:**

Strengths:
1. The identification of attractors represents a significant advancement over previous work, offering more thorough observations supported by extensive examples.
2. The study demonstrates that a variety of downstream tasks benefit from the attractors.
3. The interpretation based on an Iterated Function System (IFS) is both plausible and interpretable.

Weaknesses:
1. The paper should explicitly describe the procedure used to identify attractors. While it can be inferred that attractors are computed by averaging vectors corresponding to prompts within the same concept class, the implementation details remain vague. Many experimental settings also require further clarification. For example, in the section ``A Fractal-like Structure in the Attractors’’, some experimental setups are adopted from prior work, but the primary experimental parameters (e.g., task scope) should be explicitly stated. Without such details, readers unfamiliar with the prior literature may struggle to understand why the subtraction operation yields fractal-like structure such as -1 and -2 instead of other values.

2. The paper does not provide compelling evidence that two attractors cannot converge to similar or nearby space within the same layer. This uncertainty raises concerns about potentially modifying one attractor while retaining other concepts—particularly problematic in tasks like unlearning, where retaining out-of-scope data is crucial.

3. Although the paper proposes replacing outputs to achieve unlearning, this method may still be vulnerable to information leakage under white-box attacks.

---

> ### Author Rebuttal · Authors · 2025-07-30
>
> We thank the reviewer for their time and effort. We appreciate the insightful and interesting questions the reviewer poses. Below we will answer each one of them.
>
> ## Weakness 1: Procedure to identify attractors
>
> As correctly noted in the review, we average the latent representations of multiple inputs that fall into the same concept to estimate the attractor vector. More specifically, for Figure 4 (section 2.2) we depict different t-sne plots of the latent representations of the underlying LLM for various kinds of inputs. Each subplot demonstrates the attractor formation on different layers, for different families of concepts (e.g., layer 19 for programming languages and layer 27 for natural languages).
>
> The coloring of the points is based on the input prompts we use, e.g., orange for all JavaScript code snippets, red for all one-digit operations, and blue for all Mandarin expressions.  The layer on each subplot is chosen to be the one we identify to be the one which the inputs converge to the attractors. This can be done either in a principled manner by finding the layer with the minimal distance between embeddings, or, in a more use-case dependent way, by simply examining the behavior of each layer’s corresponding plot. In both cases, the attractor layer is extremely easy to identify as the difference with the other layers is very prominent, both distance-wise and visually.
>
> We appreciate the suggestion for the inclusion of these details explicitly, which will make the manuscript easier to understand.
>
> ## Weakness 2: Distance between attractors
>
> Thanks for the question which can help clarify the role of attractor proximity and what this means for selective concept manipulation.
>
> From an IFS standpoint, distinct concepts forming separate attractors means that the underlying contractive mappings have developed (sufficiently) different fixed point sets during pre-training. We cannot provide guarantees about minimum separation distances since this would need full characterization of the representation manifold. But the properties of IFS suggest that semantically distinct concepts should develop separated attractors. Intuitively, if two concepts consistently wanted different representational patterns during pre-training, the iterative contraction process should drive them toward distinct regions of the space.
>
> We can now provide some evidence for selective manipulation. The concern in the review was precisely what our Utility metric in the unlearning experiments (Figure 6) are designed to measure. The Utility score checks/quantifies whether modifying one concept's attractor (e.g., removing Shakespeare) affects the model's performance on related but distinct concepts (e.g., other authors or general literary knowledge). Our results show minimal utility degradation, indicating that: (a) concept attractors are sufficiently separated to allow selective manipulation, and (b) our intervention can target specific concepts without significant collateral effects on nearby concepts.
>
> Also, our experiments also indirectly test attractor separation. In the author unlearning task, we show that removing one author's influence doesn't degrade performance on other authors, despite these concepts likely occupying relatively nearby regions in the "literature" portion of the representation space. Our code translation experiments show that modifying one programming language's representation doesn't interfere with the model's understanding of other languages. These results provide empirical evidence that concept attractors maintain sufficient separation for practical applications.
>
>
> ## Weakness 3: Leakage in white-box attacks
>
> A white-box adversarial attack assumes complete access to the “internals” of a model, conditions rarely met in practical deployment scenarios.
>
> But yes, our approach, like most unlearning methods, has limited robustness against white-box adversaries with complete model access. Under such strong assumptions, an attacker could indeed perturb the attractor layer to bypass our interventions. This vulnerability is not unique to our method. Nonetheless, comprehensive adversarial robustness analysis lies outside our current scope, which focuses on establishing the core conceptual framework and demonstrating its effectiveness under standard conditions.
>
> ## Question 1: Distinctions with Knowledge Neurons
>
> This is an interesting work, and we thank the reviewer for sharing it with us which we will include.
>
> This paper is focused on knowledge neurons in encoder-based transformer models, like BERT. Knowledge neurons resemble what is also known as a “Grandmother neuron/cell” an idea from neuroscience. In this case, the assumption is that in a specific hidden layer, some of the neurons, or equivalently some of the latent vector’s dimensions, are reserved for the existence (or not) of a specific piece of knowledge. The paper is focused more on examining the extent and validity of the knowledge neurons, with experiments such as the modification of knowledge neurons and erasing.
>
> Conceptual starting point. While Knowledge Neurons is based on a neuroscientific hypothesis about localized knowledge storage, our work is grounded in dynamical systems theory. We provide a detailed framework explaining why semantic clustering emerges, whereas Knowledge Neurons primarily studies the empirical validity of neuron-level knowledge localization.
>
> Scope. Knowledge Neurons modifies specific dimensions within the latent vector (individual neurons), but our approach analyzes the geometric behavior of entire representation regions. We observe that semantically similar prompts collapse to the same attractor without neuron-level analysis or gradient computation. Our method leverages pre-existing semantic clustering patterns instead of localized neuron modifications.
>
> Architecture and use cases. Knowledge Neurons studies encoder models (BERT) for fill-in-the-blank factual knowledge tasks, whereas we study decoder-based transformers for generative applications. Our experiments describe practical applications across diverse domains from code translation to author unlearning and safety guardrails to synthetic data generation. Knowledge Neurons is interesting but was setup to study the neuroscientifically based hypothesis in more controlled settings.
> Interventions. Knowledge Neurons needs targeted neuron activation modification whereas our approach performs training-free geometric transport between pre-existing attractor regions. This makes our method more generalizable across concept types without architectural assumptions about knowledge localization.
>
>
> ## Question 2: Figure 5 - further explanation
>
> We thank the reviewer for this insightful question. In principle, an IFS can indeed exhibit non-monotonic behavior under certain conditions. For example, such behavior may arise if the IFS does not satisfy the contraction mapping requirement, or if the initial set is chosen in a way that distorts early iterations. For language models, such deviations are not just possible but likely present.
>
> However, confidently identifying non-monotonic convergence is difficult using a single visualization, such as the cosine similarity plot in Figure 5. While layer 18 may appear “closer” to the attractor structure than layer 22, this can be misleading. For instance, the boundary between the last three authors in layer 18 is far less defined than in layer 22, suggesting a more entangled representation. Instead of monotonic improvement,  we should view layers 16-24 as implementing a complex phase transition where representations undergo reorganization before stabilizing into clear attractors around layer 24. This was the purpose of Fig. 5 which we will clarify. The non-monotonic pattern may reflect the internal process of resolving competing semantic pressures.
>
> ## Question 3: Practical considerations in obtaining attractors
>
> The attractor represents a compact invariant set rather than a single point. Our averaging approach approximates the centroid of this set. But yes, a simple averaging may not be the ideal approach in forming a concept attractor. In principle, we could use a single demonstration to obtain an attractor, just like we do in Sections 4.3 and 5, since there is an online stream of data and averaging is not applicable there. Or alternatively, we could use a weighted averaging, outlier detection, etc., to mitigate what the reviewer mentions as bias to the input distribution. In our experiments, we found that simple averaging appears to work well on a broad set of experiments in the paper. We agree that this may not be the case on a different dataset or use case and will be interesting to identify situations where other sophisticated mechanisms add value.

---

> > ### Comment · Reviewer_iGkG · 2025-08-03
> >
> > I appreciate the authors' response and clarification. I will remain my positive rating.

---

### Official Review · Reviewer_Es9V · 2025-07-02

**Clarity:** 4
**Significance:** 3
**Originality:** 3
**Rating:** 4
**Confidence:** 3

**Summary:**

A novel perspective on conceptualizing Transformer-based language models is proposed. Authors start with the empirical observation that lexically different yet semantically similar inputs to LMs lead to (geometrically) close intermediate representations. A parallel between attractors in iterated function systems and in-LM concept attractors is drawn. Experiments are performed on the ability to control the model at inference time by editing such concept attractors.

**Questions:**

- I think the paper can be improved by deciding on what exact type of contribution the authors want to make. Please see the weakness section for detail.

- Please help me out in understanding the significance of section 2.1. You have formalized some parts, but never used them in the methods. Are you approximating \phi_{eff} anywhere?

- Masked language models like BERT (and their specialized variants like SentenceBERT) typically map semantically very similar but lexically very different inputs to similar dense representations. Can you extend your framework to such models?

- What about failure points (i.e., semantically very similar yet lexically different inputs mapped to different attractors due to some adversarial element)?

**Ethical Concerns:**

["NO or VERY MINOR ethics concerns only"]

**Final Justification:**

There are certain concerns that have been discussed with the authors. In good faith that the authors will perform these experiments and update the draft accordingly, I'm keeping my positive score.

**Limitations:**

Yes

**Quality:**

3

**Strengths And Weaknesses:**

## Strengths

1. *Interesting idea*: The possibility of approximating an LM using iterated functions is intriguing. It can be used as an alternate way of interpreting these models. In general, I am inclined to functional decomposition of a model, instead of structural.

2. *Causal importance* I appreciate that the authors went on to actually showcase the causal viability of these structures that they identify. Many interpretability works stop at identifying the structure; however, without actually showing the effects of editing such structures, the scope becomes limited.


## Weakness:

The primary weakness of this paper is in its positioning of contribution. Is it about a novel perspective on interpreting LMs? Then, one would expect a more in-depth positioning of this perspective against existing techniques/conceptualization: where do they fail, what do they miss, how does the proposed approach capture it? Additionally, the paper should discuss different existing works looking at Transformers from a dynamical system perspective [1, 2, 3].

Instead, if this paper is a new method of steering (like all the applications experimented with), then it needs to discuss the limitations of the SoTA (e.g., [4] and [5] for example), experiment with existing such methods and show that the attractor-based steering is better.

---

> ### Author Rebuttal · Authors · 2025-07-30
>
> We appreciate the time and effort put in by the reviewer. Below we analyze in detail each question that the reviewer has.
>
> ## Weakness/Question 1: Paper’s contribution/positioning
>
> We appreciate the reviewer's constructive feedback on positioning. Briefly, we consider the contribution to cover both interpretability and five distinct practical applications through a unified framework.
>
> IFS: Our core contribution shows that LLMs appear to implement concept-specific Iterated Function Systems, where semantic clustering emerges from contractive dynamics across layers. This framework, appreciated by all reviewers, unifies previously disconnected observations: (a) belief state representations emerge as invariant measures over our concept attractors, (b) their observed fractal structures arise directly from IFS self-similarity properties, and (c) the attractor dynamics represent the convergent trajectories we formalize mathematically. Unlike existing approaches that identify discrete circuits without getting much into underlying causes, we give a clear machinery to understand why semantic representational structures emerge and how they can be systematically manipulated.
>
> Intervention Strategy: The IFS setup gives a generic intervention method that shows that different applications can be approached via a single mathematical principle. We do not develop separate strategies for code translation, author unlearning, safety guardrails, and synthetic data generation. Our approach gives a common approach for all these tasks. We hope you agree that this is a useful standalone result relative to existing steering methods like representation adjustment or activation patching. We use pre-existing geometric structure, making interventions principled and efficient.
>
> Methods: Our paper suggests that we can shift from tailored interventions to principled geometric reasoning about LLMs. Instead of treating steering, safety, and generation as separate engineering challenges, we can potentially approach these problems through a unified attractor manipulation idea. We show that IFS can be successfully recovered, so the use-cases we show are meaningful manipulations of learned representational geometry rather than purely empirical ideas that work well.
>
> ## Question 2: Significance of Section 2.1
>
> Section 2.1 provides the technical foundation that directly enables all of our practically focused methods in Sections 4-6. Yes, while we do not explicitly compute $\phi_{eff}$, the framework shows why our entire approach is possible.
>
> Consider a few examples. Our "teleportation" operations that add attractor difference vectors are approximations of the complex IFS dynamics $\phi_{eff}$ represents. Without this grounding, any operations (like vector adjustments) would appear ad-hoc; with it, these operations become meaningful transport maps between attractor basins. The invariant measure $\mu_C$ formalized in Section 2.1 directly explains why our interventions can be expected to maintain some semantic coherence during generation: when we teleport representations, subsequent generation will sample from the target concept's invariant measure, so this ensures consistent outputs across applications. The optimization model we solve for IFS recovery minimizes Wasserstein distance under contraction constraints: this is approximating the Collage Theorem machinery from Section 2.1. This shows that real LLMs exhibit the hypothesized IFS properties, providing empirical support for $\phi_{eff}$. Rather than requiring computationally prohibitive explicit $\phi_{eff}$ calculation, we use a simple approximation. Overall, the purpose of the section is to setup the theoretical anchor points, so that the reader can appreciate how these would then inform  practical algorithm development.
>
>
>
> ## Question 3: Application to BERT
>
> This is an excellent question. While in principle, IFS should apply in some form to encoder models, there are some challenges.
>
> The bidirectional attention mechanism will create different contraction patterns compared to autoregressive models, but the principle of semantic clustering through layer transformations should hold. The success of sentence embeddings from BERT variants in capturing semantic similarity gives preliminary evidence that some type of concept-like clustering should be occurring in these models. But extending to BERT would require non-trivial changes to both our analysis and applications. The bidirectional processing means concept attractors would likely be distributed across token positions rather than concentrated in final tokens. Additionally, our interventions  would need to be redesigned since BERT doesn't perform autoregressive generation (modified representations influences subsequent tokens).
>
> This paper focuses on autoregressive models because (a) The applications (code translation, text generation, synthetic data creation) need generative capabilities that encoder models don't directly offer, (2) Semantic collapse we use is more directly observable in autoregressive models where information accumulates unidirectionally, and (3) Our interventions described in the paper are well-suited for models that continue generation from modified representations.
>
> Our goal was to describe the framework in the autoregressive setting where the theoretical guidance is clearest and the practical applications quite natural. It gives a good starting point for extensions to masked language models.
>
>
> ## Question 4: Failure points
>
> This is a very interesting question, which we will incorporate in the paper with credit to the reviewer.
>
> While we did not encounter systematic or catastropic failures in our experiments, we can recognize several potential failure cases: (a) concepts where words legitimately belong to multiple semantic categories may not form coherent single attractors and cross-domain concepts that span multiple conceptual boundaries may exhibit unstable attractor membership, and (b) under-represented concepts with insufficient training data may also not develop stable attractors.
>
> The scenario of deliberately crafted inputs designed to exploit attractor boundaries is a genuine vulnerability. Such a prompt could indeed combine surface features from one concept with semantic content from another, causing mis-attribution to the wrong attractor. We will include this text in the paper.
>
> In closing, we should distinguish between two types of failures. The first is the intrinsic framework limitations where our IFS formulation fails to capture the model's actual representational structure, and the second, model-inherited limitations where the underlying LLM's learned representations are themselves inconsistent or exploitable. The adversarial scenario falls primarily into the second category, and yes, our framework will inherit these weaknesses rather than correct them.

---

> > ### Comment · Reviewer_Es9V · 2025-08-04
> >
> > Thank you for your detailed response. Unfortunately, these do not address my specific concerns. On top of that, the provided response adds to some new confusion:
> >
> > If you are not computing the IFS, are you computing the attractors by averaging across prompts? Then, how is it different from standard steering vectors, and how is it free from the limitations of these steering vector-based approaches?

---

> > > ### Author Response · Authors · 2025-08-05
> > >
> > > Thanks for pushing us to explicitly pin down the difference. The main distinction between our attractor centroids and standard steering vectors lies in what the averaging represents and the technical framework that enables it.
> > >
> > > 1. First, when we average concept-related prompts, we are not arbitrarily combining vectors. We are approximating the centroid of a mathematically predicted invariant set. If the IFS hypothesis holds (we have extensive empirical evidence supporting it), it predicts that semantically related prompts should converge to the same geometric region (the attractor). The averaging operation gives us a summary of this geometric structure; we are not simply constructing an empirical steering direction. This summary, derived by averaging, works immediately across diverse concept types (fictional worlds, programming languages, authorial styles) without task-specific tuning. We can appreciate that we are discovering pre-existing geometric organization rather than creating steering directions to address the needs of a specific use case.
> > >
> > > 2. From a practical standpoint, standard steering methods mostly require carefully constructed contrast pairs and/or training of auxiliary models, as we explain in lines 229-232. This is because they lack concrete guidance about where (or if) semantic structure should emerge. Our IFS framework takes this as a starting point. It tells us exactly how to look and what to expect (concept clustering), and averaging is simply to keep things simple and works surprisingly well. We hope you agree that the technical framing transforms what would otherwise be an ad-hoc heuristic into a mostly principled approximation.
> > >
> > > 3. While averaging gives an effective initial approximation, our formulation does suggest more interesting options, based on proximity to attractor boundaries, multi-centroid representations for concepts with multiple attractors, etc. The IFS setup lays out a coherent plan for further development rather than purely empirically guided enhancements to steering methods (to address shortcomings).
> > >
> > > 4. Finally, our construction works across very different applications—code translation, author unlearning, safety guardrails -- using the same operations, because the IFS framework reveals that these are all instances of transport between concept attractors. Standard steering methods require separate engineering for each domain because they lack this unifying principle.

---

> > > > ### Comment · Reviewer_Es9V · 2025-08-05
> > > > **Necessity of data**
> > > >
> > > > > While averaging gives an effective initial approximation, our formulation does suggest more interesting options, based on proximity to attractor boundaries, multi-centroid representations for concepts with multiple attractors, etc. The IFS setup lays out a coherent plan for further development rather than purely empirically guided enhancements to steering methods (to address shortcomings).
> > > >
> > > > > From a practical standpoint, standard steering methods mostly require carefully constructed contrast pairs and/or training of auxiliary models, as we explain in lines 229-232. This is because they lack concrete guidance about where (or if) semantic structure should emerge.
> > > >
> > > > (Please correct me if I'm wrong) I suppose you cannot locate the attractors (and the concepts) in closed form? That is, if I take the TOFU dataset as example, you need to compute the author-specifc attractors from a bunch of prompts related to the author. Similarly, for detoxification, you would need carefully chosen prompts to compute the attractors that correspond to the 'toxic' attractors and 'non-toxic' attractors. Then, how can you claim the non-necessity of carefully chosen data points? Does your method automatically guides the data sampler to look for specific data?
> > > >
> > > > Currently, it looks like the only difference between your method and additive steering is that you define the source and the target points of steering, while the latter defines the direction (which is, difference between the source and target).

---

> > > > > ### Author Response · Authors · 2025-08-06
> > > > >
> > > > > Thanks for engaging with us! You are correct that we cannot locate attractors in closed form and do require some concept-related prompts. However, two main distinctions are advantageous over steering approaches directly (ignoring conceptual differences for now).
> > > > >
> > > > > Standard steering methods require carefully set up contrast pairs (toxic versus non-toxic examples, honest versus dishonest responses). This gives directional differences. These pairs must be semantically matched except for the target attribute. We know from existing work on steering that this needs curation.
> > > > > **Our method needs only some positive examples of each concept (Python code samples, Shakespeare text, Harry Potter references). No contrastive pairing or semantic matching is needed.** Just representative examples of the concept suffice in all our experiments (Figure 7).
> > > > >
> > > > > Also, steering methods cannot easily address many of our use cases:
> > > > >
> > > > > 1. For code translation, steering needs contrast pairs, but collecting "non-Python" vs "Python" contrast for translating algorithms between languages is not straightforward. In our setup, we only need to identify each language's attractor independently.
> > > > >
> > > > > 2. For author unlearning, some steering-type experiments would need "Shakespeare versus non-Shakespeare" pairs, but "non-Shakespeare" is not a very coherent concept. Our method removes Shakespeare's attractor without needing contrastive definitions.
> > > > >
> > > > > 3. For synthetic data generation, steering has no easy mechanism for generating new content within concept boundaries. Our method samples from attractor regions to create novel examples.
> > > > >
> > > > > These differences come directly from components that we have covered in our discussion with you, for which we are grateful. Steering computes directions between contrasts, while we identify absolute locations in representation space. IFS tells us that these concept-specific locations exist independently; we approximate them through averaging. This allows applications that are difficult with directional steering approaches.
> > > > >
> > > > > To summarize, for detoxification and code translation (the only steering-related applications), standard methods need paired examples like "I hate this person" versus "I dislike this person's behavior." We **only** need examples of toxic content; no pairing is required.
> > > > >
> > > > > The reviewer correctly notes both methods perform vector arithmetic, but we allow a much broader class of applications that steering approaches cannot easily address by design. We will be permitted one additional page and we can absorb much of this discussion into the paper. Thanks again!

---

> ### Comment · Reviewer_Es9V · 2025-08-07
> **Empirical justification might improve the argumentation**
>
> Thanks for your clarification!
>
> I do agree with the intuitive argumentation. However, empirical validation would be needed to justify this.
>
> For example, you can establish the relative superiority of the attractor-based steering over additive steering by showing that:
>
> 1) The examples that are used for attractor-based steering (successfully) is not enough if one tries additive steering;
>
> 2) With the carefully curated examples of standard steering methods, attractor-based steering yields superior results;
>
> 3) Maybe demonstrate that the attractor-based steering is free from (or less affected by) the unreliable behaviors of steering vectors, e.g. characterized by [1];
>
> 4) Maybe evaluate over some existing benchmarks like Axbench [2];
>
> My precise opinion is, even methods with robust theoretical proofs require empirical justification to establish their claims. In its current form, the concept attractor-based steering, however intuitively appealing they are, needs to be empirically validated against existing steering methods (of course not the ones that require training, but those that use algebraic solutions).
>
>
> [1] Braun et al., Understanding (Un)Reliability of Steering Vectors in Language Models, 2025
>
> [2] Wu et al., AxBench: Steering LLMs? Even Simple Baselines Outperform Sparse Autoencoders, 2025

---

> > ### Author Response · Authors · 2025-08-07
> > **really appreciate the suggestion!**
> >
> > Dear Reviewer Es9V,
> >
> > We much appreciate your constructive engagement and acknowledging the potential benefits of our framework. Your suggestion for additional empirical validation for steering in particular also aligns with our goal of showing practical benefits together with technical contributions.
> >
> > Scope/Initial focus in the submission: In the paper, we intended to introduce the IFS framework and demonstrates its broad uses across diverse applications, we are grateful that this was appreciated. Steering was one application among several. Our goal was to show that attractor-based methods can achieve comparable performance.
> >
> > However, we welcome your suggestion to conduct deeper empirical study for steering. With the one page allowance, we will incorporate a dedicated steering comparison section that addresses the reliability issues highlighted in Braun (2025) and benchmarks in Wu (2025). This analysis will check: (1) whether attractor-based steering exhibits the high variance and anti-steering behavior, and (2) check performance on AxBench tasks, and (3) analyze different prompt types and concept boundaries.
> >
> > Outcome? Whether attractors outperforms, matches, or underperforms standard steering, the results will be useful. If performance is comparable, our paper will show that attractor-based methods offer similar effectiveness with reduced data requirements. If we identify cases where our approach shows clear superiority, this would give a nice contribution, making it more appealing to many colleagues working in steering.
> >
> > Thank you for these constructive suggestions, general encouragement of the work and the recent references, which we will include.

---

### Official Review · Reviewer_Rwmy · 2025-07-03

**Clarity:** 3
**Significance:** 3
**Originality:** 3
**Rating:** 4
**Confidence:** 3

**Summary:**

The paper identifies that semantically similar prompts, regardless of their surface text differences, converge to similar internal representations within specific layers of Large Language Models (LLMs). It proposes that this convergence behavior can be understood through the lens of Iterated Function Systems (IFS). Specifically, it views successive layers of the LLM as applying contractive mappings that iteratively drive representations towards stable, concept-specific points called Attractors. Leveraging this IFS/Attractor theory, the authors develop simple, training-free techniques for various NLP tasks by manipulating these Attractors.

**Questions:**

See Weakness

**Ethical Concerns:**

["NO or VERY MINOR ethics concerns only"]

**Final Justification:**

All my concerns arre addressed and I decide to give a positive score.

**Limitations:**

Yes

**Quality:**

3

**Strengths And Weaknesses:**

Strengths：
1. This paper explores an interesting topic.
2. The paper is well-written.
3. Experiments around the application of concept attractors are comprehensive.

Weaknesses：
1. The definition of concept is unclear. What exactly is a concept? When discussing things at a macro level, we might consider C++ and Python as belonging to the same concept—both being programming languages. However, when delving into more specific details, C++ and Python could represent different concepts.
2. Beyond applications, further discussion centers on how concept attractors are formed. How do training procedures and model architectures influence their formation? Would supervised fine-tuning (SFT) alter these attractors?

---

> ### Author Rebuttal · Authors · 2025-07-30
>
> We thank the reviewer for their time and effort. Below, we analyze each of their questions. We hope this will help clarify any remaining concerns about our work and strengthen the support for it.
>
> ## Weakness 1: Concepts on various levels; definition of concept unclear
>
> This is an excellent point and gets to our formulation’s core. We want to clarify that the flexible definition of "concept" is a strength that makes the idea broadly applicable across diverse use cases.
>
> The main question is “what is a concept?”. In the paper, a concept is defined through a data-driven process rather than arbitrary categorization. As you noted, the same entities (C++ and Python) can legitimately represent either the same concept or different concepts depending on the task objective, and our construction empirically validates whether such distinctions are meaningful in the LLM's representational space. Let us check the process briefly. First, we gather prompt sets based on task requirements (e.g., separate collections for C++ and Python in code translation, or merged collections if they are one concept). Then, we process these prompts and identify the layer where representations cluster most tightly. The LLM's dynamics determine whether the conceptual distinction is meaningful—if prompts do not form distinct attractors, the proposed concept boundary is not supported by the model's representations.
>
> We can check specific examples. In code translation (section 4.2), C++ and Python form distinct attractors at layer 19, as we see from Figure 4, because the task requires translation between languages, and the LLM has learned to represent them distinctly. In author unlearning (section 3) individual authors form separate attractors because we want author-specific representational patterns. If we had instead collected mixed programming language prompts and checked if they clustered together, this would indicate a shared "programming" concept at that layer. What is interesting is that rather than imposing predetermined conceptual hierarchies, our framework lets the LLM's learned representational structure determine meaningful concept boundaries depending on the use case. Your observation about hierarchical concept levels is absolutely correct, and our framework naturally lets us discover these hierarchies through layer-specific attractor formation (e.g., Figure 4 left). This is both flexible for diverse applications and allows adjustment  based on data as needed.
>
> If the concern is about reproducibility, we assure the reviewer that the full codebase to replicate the experiments with the flexibility to extend the current setup to novel concepts based on specifying a bag of prompts will be available with the paper.
>
>
>
> ## Weakness 2: How model architecture influences the attractors
>
> Thanks for this question. While a full analysis across all many architectures was not within the scope of this initial work, we can provide some insights based on our observations.
>
> Our experiments show that concept attractors form consistently across different LLM architectures within the same family (Llama variants). This  leads us to suspect that attractor formation is not an artifact of architectural choices but emerges from transformer layers acting as iterative contractive mappings.
>
> Regarding fine-tuning, from an IFS perspective, attractor formation is based on the existence of contractive mappings that preserve semantic relationships. So, as long as fine-tuning (or training) gives representations where semantically related content clusters, the key conditions for attractor formation remain satisfied. This suggests that attractors should persist across different fine-tuning/training procedures as long as semantic coherence is maintained. This mostly aligns with findings in the literature on fine-tuning effects involving task vectors and representation analysis, which show that fine-tuning typically modifies rather than destroys the semantic organization of the language model.

---

> > ### Comment · Area_Chair_sJmq · 2025-08-07
> > **Please Engage in Author Response Discussion**
> >
> > Hi Reviewer Rwmy,
> >
> > We encourage you to review the authors’ rebuttals and see how they’ve addressed your comments. If you’ve already done so, thank you! Kindly confirm your engagement by reacting in this thread.
> >
> > Your participation helps ensure a fair and thoughtful review process.
> >
> > Best regards, AC

---

> > ### Comment · Reviewer_Rwmy · 2025-08-08
> >
> > I appreciate the authors' response and clarification. My concers are addressed. I will be updating my scores.

---

### Official Review · Reviewer_XtJn · 2025-07-05

**Clarity:** 3
**Significance:** 3
**Originality:** 4
**Rating:** 4
**Confidence:** 4

**Summary:**

The paper takes inspiration from the notion of attractors in IFS and uses the idea in the concept space or embedding space. A variety of anecdotal evidence on induced tokens which are semantically relevant, layer sensitivity of attractors, multiple attractors for the same concept, and training-free hallucination detection, sanitizing toxic text, transpiling programming languages, improving vision-language models, and high-quality synthetic data generation. The IFS itself is learned to minimize the difference between the IFS output when started on the lowest layer and the highest layer output.

**Questions:**

1.	The paper mentions fractal like nature of attractors. Have you measured fractal dimensions of these sets?

2.	Provide details of the training procedure outlined in lines 103-133.

3.	How are attractors shown in Figure 4 identified and labeled?

4.	How does “teleportation” of attractors mentioned in Sections 4.1 and 4.3 performed?

**Ethical Concerns:**

["NO or VERY MINOR ethics concerns only"]

**Final Justification:**

One source of confusion was that I had missed the appendix. It is resolved now.

The authors have answered questions and clarified the issues.

I am increasing the score from 2 to 4.

**Limitations:**

Yes

**Paper Formatting Concerns:**

No formatting issues.

The main paper is not self-contained without the appendixes/

**Quality:**

3

**Strengths And Weaknesses:**

S1. The idea of concept attractors is intriguing and is capable of bringing new insights to the functioning of LLMs. The paper gets credit for connecting the two ideas.

S2. The approach is demonstrated on a wide range of tasks.

W1. The biggest weakness is that there are no appendixes in downloaded version. Details of attractor framework, which is central to the paper, as well as the details on experiments are missing. My scores are mainly due to lack of information.

W2. The main paper is not self-contained as the main points are explained in the appendixes.

W3. The idea needs access to internals of LLM. This is a minor issue as the benefits outweigh. This is also called out in the paper.

---

> ### Author Rebuttal · Authors · 2025-07-30
>
> Dear Reviewer XtJn,
>
> Thanks for the time you’ve invested in our paper. Before answering the specific questions one by one, we want to address your main concern: **the lack of an appendix**. A detailed appendix with the dataset details, the metrics used, and an extensive set of additional results are available in the “supplementary” material we uploaded with the paper, per the conference’s guidelines. If you locate it, we are confident that it will clarify most of the high-level missing details concerns about our work.
>
> ## Question 1: Fractal dimension of sets
>
> While the "fractal-like" structure we observe is a technical anchor that drives our development and use in the variety of applications that all reviews found valuable, we did not attempt to measure the fractal dimension for the following reasons, based on which you will see is computationally/statistically intractable in the context of most models like Llama.
>
> 1. Standard algorithms for fractal dimension estimation (e.g., box-counting) are designed for low-dimensions. Running these methods is difficult because partitioning the space with hyper-cubes (in box-counting) is very intensive and, further, must also avoid singularities like data points being isolated in their own partition. This is infeasible even if we had unrealistically large sample sizes (see point #2 below).
> 2. Sample size needs for these methods to achieve a sensible result with a grid resolution of $\epsilon$ is $O(\epsilon^{-d})$. This is extremely large to generate or even store for feeding into an algorithm.
>
> Of course, our concept attractors are almost certainly low-dimensional manifolds embedded in the high-dimensional space of LLM activations. The difficulty is that classical fractal dimension estimators are not designed to disentangle the properties of the embedding from the intrinsic geometry of the manifold itself. So, estimating the intrinsic fractal properties of the concept set via classical methods is difficult to even attempt.
>
> For this reason, the fractal dimension, which is theoretically interesting, is practically out of reach. So, we focused on the conceptual formulation and the more tractable geometric/t-SNE type properties that directly support the paper's main contributions.
>
> ## Question 2: Training procedure for estimating the IFS
>
> The training procedure is described in lines 103-120. Briefly, we try to find the best contrastive mapping $M_{eff}$ which minimizes the difference between the embeddings of layer $l$ and layer $0$ (input embeddings). We apologize if this is not clear from our description as we realize there is a typo in line 115 (Figure 2 instead of 3). We will update the manuscript accordingly.
>
> ## Question 3: Further explanation of Figure 4
>
> Similar to Figure 1, we do not perform *any* processing of the raw output of the LLM. What we show in Figure 4 are different t-sne plots of the latent representations of the underlying LLM for various kinds of inputs. Each subplot demonstrates the attractor formation on different layers, for different families of concepts (e.g., layer 19 for programming languages and layer 27 for natural languages).
>
> The coloring of the points is based on the input prompts we use, e.g., orange for all JavaScript code snippets, red for all one-digit operations, and blue for all Mandarin expressions.  The layer on each subplot is chosen to be the one we identify to be the one that the inputs converge to the attractors. This can be done either in a principled manner by finding the layer with the minimal distance between embeddings, or, in a more use-case dependent way, by simply examining the behavior of each layer’s corresponding plot. In both cases, the attractor layer is extremely easy to identify as the difference with the other layers is very prominent, both distance-wise and visually.
>
> ## Question 4: Teleportation explanation in Sections 4.1 and 4.3
>
> The teleportation operation means a controlled perturbation of LLM's representation space to redirect generation toward a target concept's attractor. We do this as follows:
>
> Given a representation $h\_l(p)$ at layer $l$ that lies within/near the attractor  $A\_{\mathcal{C}\_1}$ of concept $\mathcal{C}\_1$, and a target attractor $A\_{\mathcal{C}\_2}$ for concept $\mathcal{C}\_2$, we can transform:
>
> $$h\_l'(p) = h\_l(p) + \alpha\bar{A}\_{\mathcal{C}\_2}$$
>
> where $\bar{A}\_{\mathcal{C}\_2}$ is the centroid of attractor $A\_{\mathcal{C}\_2}$ and $\alpha$ is a scaling parameter.
>
> This operation is simply doing an instantaneous transport map between attractor basins. Notice that each attractor $A\_{\mathcal{C}}$ represents the fixed point set of concept-specific contractions. The newly introduced vector $(\bar{A}\_{\mathcal{C}\_2})$ captures the translation needed to move from the geometric center of one fixed point set to another. Adding this vector to a representation effectively moves it to the basin of attraction for the target concept, where later layer transformations will naturally contract it toward $A\_{\mathcal{C}\_2}$.
>
> In practice we:
>
> 1. Identify the attractor layer $l\_{\mathcal{C}}$ for each concept, by identifying the layer that the embeddings converge to specific points (the attractors).
> 2. Compute the attractor centroids by averaging representations of multiple concept-related prompts: $\bar{A}\_{\mathcal{C}} = \frac{1}{n}\sum_{i=1}^n h_{l\_\mathcal{C}}(p_i^{\mathcal{C}})$.
> 3. During inference, intercept/interrupt the representation at layer $l_\mathcal{C}$ and apply the transport transformation
> 4. Allow subsequent layers to process the modified representation normally
>
> The process is quite simple. Unlike approaches requiring auxiliary model training, we can use the pre-existing attractor structure. This removes the need for concept-specific datasets, training procedures, or model parameter modifications by using the LLM's inherent representational geometry.
>
> A discussion and explanation of the above is in lines 219-240 and Figure 7. All three subsections (4.1 through 4.3) follow the same setup but we are happy to expand it further.

---

> > ### Comment · Reviewer_XtJn · 2025-08-04
> > **Apologies and Thanks**
> >
> > My apologies for missing the supplementary section.  I had downloaded the zip file with all the papers to be reviewed. In the past, this zip file had supplementary material but not this time. In addition, a few other papers had supplementary content in the main pdf itself. My apologies again.
> >
> > Thanks for responding to the comments. I have follow-up questions.
> >
> > 1. As part of response to 'Question 2' above, can you please share a crisp and clear description of the of training procedure?
> >
> > 2. In your response to 'Question 4' above, does the centroid of $\mathcal{C}_1$ play a role in the transformation? The expressions seems incomplete.
> >
> > Regarding the use of fractal dimension in the paper: I do not think mentioning a mathematical term without any measurement is justified. It is at best a hypothesis. My recommendation is to drop the use of the term 'fractal' in the paper and add a footnote about the hypothesis. Strong statements such as "A fractal-like structure in the Attractors" require measurement of fractal dimension, in my view.

---

> > > ### Author Response · Authors · 2025-08-05
> > >
> > > We sincerely appreciate your continued engagement, which we know is time-consuming. We are glad to have the chance to clarify some of these details. Thanks for locating the supplement!
> > > ### Question 2 algorithm
> > > The procedure approximates the Collage Theorem to recover concept-specific contractions. We alternate between two steps: (1) fit contraction mappings that minimize the distance (e.g. Wasserstein or some other suitable distance) between source representations and target attractors, subject to spectral norm constraints ensuring contractivity, and (2) reassigning vector pairs to their best-fitting contraction map. The algorithm implements this as follows: fit_contraction optimizes individual linear maps $A$ with spectral norm to minimize $||AX - Y||^2$, while alternating_contraction_fits clusters vector pairs and iteratively refines both cluster assignments and their corresponding contraction maps. This operationalizes the framework from Section 2.1, providing empirical validation that real LLMs appear to exhibit IFS like properties. The spectral norm constraint allows each learned mapping to satisfy the definition of a contraction.
> > >
> > > ```python
> > > def spectral_norm_projection(A, rho=0.99):
> > > 	"""Project matrix A onto the spectral norm ball of radius rho, robust to SVD failures."""
> > > 	try:
> > > 		U, S, Vt = svd(A, full_matrices=False)
> > > 		S_clipped = np.minimum(S, rho)
> > > 		return U @ np.diag(S_clipped) @ Vt
> > > 	except np.linalg.LinAlgError:
> > > 		# Fallback: return scaled-down A
> > > 		A_norm = norm(A, 2)
> > > 		if A_norm == 0:
> > > 			return A
> > > 		scale = min(rho / A_norm, 1.0)
> > > 		return A * scale
> > >
> > > def fit_contraction(X, Y, rho=0.99, max_iter=100, lr=1e-2):
> > > 	"""Fit a linear map A with spectral norm < rho to minimize ||AX - Y||^2."""
> > > 	d = X.shape[1]
> > > 	A = np.random.randn(d, d) * 0.01  # Small random init
> > > 	for _ in tqdm(range(max_iter)):
> > > 		grad = 2 * (A @ X.T - Y.T) @ X  # Gradient of ||AX - Y||^2
> > > 		A -= lr * grad
> > > 		A = spectral_norm_projection(A, rho)  # Ensure contraction
> > > 	return A
> > >
> > > def alternating_contraction_fitting(X, Y, k=3, rho=0.99, max_iter=100):
> > > 	"""
> > > 	Cluster vector pairs into k groups, each with its own contraction map A_j,
> > > 	such that A_j approximately maps all x_i in its group to corresponding y_i.
> > > 	Args:
> > > 		X: array of shape (n, d), source vectors
> > > 		Y: array of shape (n, d), target vectors
> > > 		k: number of contraction maps
> > > 		rho: upper bound on spectral norm (should be < 1)
> > > 		max_iter: number of alternating optimization rounds
> > > 	Returns:
> > > 		A_list: list of fitted contraction matrices A_j
> > > 		sigma: assignment array of shape (n,), indicating which A_j is used for each pair
> > > 	"""
> > >
> > > 	n, d = X.shape
> > > 	sigma = np.random.choice(k, size=n)  # initial random assignments
> > > 	A_list = [np.eye(d) * 0.01 for _ in range(k)]  # initial small maps
> > >
> > > 	# Step 1: Fit maps A_j to each cluster
> > > 	for j in range(k):
> > > 		A_list[j] = fit_contraction(X, Y, rho=rho, max_iter=max_iter)
> > >
> > > 	# Step 2: Reassign each pair to the best fitting map
> > > 	errors = np.zeros((n, k))
> > > 	for j in range(k):
> > > 		AX = X @ A_list[j].T
> > > 		errors[:, j] = np.sum((AX - Y) ** 2, axis=1)
> > > 		sigma = np.argmin(errors, axis=1)
> > >
> > > 	return A_list, sigma
> > > ```
> > >
> > > ### Need for source centroid: clarification
> > >
> > > We should clarify the expression: the theoretical transformation $h_l'(p) = h_l(p) + \alpha (\bar{A}\_{C\_2} - \bar{A}\_{C\_1})$ does not require the source centroid $\bar{A}\_{C\_1}$ for our practical implementation. This is because in our experiments (Section 4, Figure 7), we implement a simplified version: $h_l'(p) = h_l(p) + \alpha \cdot \bar{A}\_{C\_2}$, where we add only a nudge towards the target attractor centroid. This works because adding the target attractor vector creates sufficient bias toward the target concept's basin of attraction, regardless of the source representation's original position. The full expression $h_l'(p) = h_l(p) + \alpha(\bar{A}\_{C\_2} - \bar{A}\_{C\_1})$ gives the complete transport map between attractor centroids, but we find that in practice, we can get strong results with just the target addition. In fact, this is one of the main strengths of our algorithm compared to other steering baselines (as mentioned in lines 229-232), as it requires no paired data or even data from the source concept. We will clarify and apologize for the confusion.
> > >
> > > ## Fractal usage
> > >
> > > We appreciate the suggestion and we will explicitly state that this is a hypothesis following the theoretical properties of IFS, and not a formal definition of a fractal structure. Much appreciate the suggestion!
> > >
> > > We hope that these clarifications and the content that was provided in the supplement have addressed your concerns. We are happy to answer any additional questions!

---

> > > > ### Comment · Reviewer_XtJn · 2025-08-07
> > > > **Thanks**
> > > >
> > > > Thanks. I will be updating my scores.

---

### Note · Authors · 2025-08-12

We thank all reviewers and the Area Chair for their time and effort. All the reviewers engaged in discussions, appreciated the many benefits/novelty of our formulation, and unanimously recommended acceptance. We are grateful for the insights/suggestions in our discussion, and all of this will be incorporated in our final, published version.

---

### Decision · Program_Chairs · 2025-09-17

**Decision:**

Reject

**Comment:**

This paper shows that semantically similar prompts converge to similar internal representations within layers of LLMs, despite differences in their text surfaces. The authors interpret this phenomenon as Iterated Function Systems and Attractors, and propose simple techniques to manipulate these Attractors for downstream tasks.

Overall, the reviewers acknowledged that the main finding both interesting and intriguing. However, an important concern remains regarding the positioning of the contributions; whether to the interpretability or the methodology for leveraging Attractors, which resulted in insufficient analysis/experiments. These issues constrain the strength and validity of the work in its present form.

I encourage the authors to carefully address the corresponding reviewer’s comments, expand the experimental evaluation, and clarify the scope of the contributions. This is a promising direction, and with further development, it has the potential to lead to a more substantial impact in future work.